# Emergence of superconductivity in the cuprates via a universal percolation process

Damjan Pelc [1,5], Marija Vučković[1,6], Mihael S. Grbić [1], Miroslav Požek [1], Guichuan Yu[2], Takao Sasagawa[3], Martin Greven[2] & Neven Barišić[2,4]

A pivotal step toward understanding unconventional superconductors would be to decipher how superconductivity emerges from the unusual normal state. In the cuprates, traces of superconducting pairing appear above the macroscopic transition temperature $T_c$, yet extensive investigation has led to disparate conclusions. The main difficulty has been to separate superconducting contributions from complex normal-state behaviour. Here we avoid this problem by measuring nonlinear conductivity, an observable that is zero in the normal state. We uncover for several representative cuprates that the nonlinear conductivity vanishes exponentially above $T_c$, both with temperature and magnetic field, and exhibits temperature-scaling characterized by a universal scale $\Xi_0$. Attempts to model the response with standard Ginzburg-Landau theory are systematically unsuccessful. Instead, our findings are captured by a simple percolation model that also explains other properties of the cuprates. We thus resolve a long-standing conundrum by showing that the superconducting precursor in the cuprates is strongly affected by intrinsic inhomogeneity.

[1] Department of Physics, Faculty of Science, University of Zagreb, Bijenička 32, HR-10000 Zagreb, Croatia. [2] School of Physics and Astronomy, University of Minnesota, Minneapolis, MN 55455, USA. [3] Materials and Structures Laboratory, Tokyo Institute of Technology, Kanagawa 226-8503, Japan. [4] Institute of Solid State Physics, TU Wien 1040 Vienna, Austria. [5] Present address: School of Physics and Astronomy, University of Minnesota, Minneapolis, MN 55455, USA. [6] Present address: University Hospital Centre Zagreb, Kišpatićeva 12, HR-10000 Zagreb, Croatia. These authors contributed equally: Damjan Pelc, Marija Vučković. Correspondence and requests for materials should be addressed to M.Pže. (email: mpozek@phy.hr) or to M.G. (email: greven@umn.edu) or to N.Bšić. (email: neven.barisic@tuwien.ac.at)

Despite tremendous experimental and theoretical efforts over the past three decades, the nature of the superconducting fluctuation regime of the cuprates remains intensely debated[1]. Experimentally, the problem has been approached using bulk probes, such as conductivity and other transport properties in a wide frequency range[2–13], magnetic susceptibility[3,14–16], surface sensitive probes[17], local probes, such as muon spin rotation[18], and photoemission spectroscopy[19,20]. Some studies point to the possible persistence of superconducting pairing well above $T_c$, which has been taken as an indication of preformed Cooper pairs related to the appearance of the pseudogap[13,21]. Other studies indicate that traces of superconductivity emerge at somewhat lower temperatures, and are most prominent at moderate doping[9,10,14]. High pairing onset temperatures have been related to exotic normal-state physics[22,23] and to unconventional prepairing[24,25], with profound consequences for the mechanism of cuprate superconductivity. However, terahertz and microwave conductivity[2,5,6,11,12] as well as magnetometry experiments[15,16] consistently detect superconducting contributions only near $T_c$, irrespective of doping. The resolution of this puzzle would be a crucial step toward understanding the high-$T_c$ cuprates.

However, in previous experiments it has often been difficult to reliably establish the nonsuperconducting normal-state contribution in order to extract a superconducting signal. Typical approaches involve the extrapolation of high-temperature behavior or the suppression of superconductivity by a magnetic field. The situation is further convoluted due to the complexity of the cuprate phase diagram, which features a doping-dependent pseudogap, as well as universal and compound-specific ordering tendencies that manifest themselves differently in different experimental observables[1]. The presence of various kinds of disorder in these complex oxides poses yet another complication[26]. Data are often discussed assuming preformed Cooper pairs in an extended temperature range[9,10,14], or analyzed within the Ginzburg–Landau (GL) framework with possible corrections to the original mean-field theory[3,4,7,8,12]; yet this has not resulted in a unified picture.

The absence of any discernible signal due to nonsuperconducting contributions renders the nonlinear conductivity technique uniquely suitable to study and model superconductivity emergence. We apply this probe to a number of cuprate families and a variety of experimental conditions. The measurements unambiguously show that the superconducting precursor is limited to a narrow temperature range above $T_c$, which rules out extended fluctuations and prepairing regimes. Importantly, we find that the superconductivity emergence range is not controlled by $T_c$, a crucial qualitative feature of GL theory, but rather by a scale $\Xi_0$ that is nearly independent of compound and doping (in the studied doping range $p = 0.08–0.19$). This robust experimental finding is an important step toward understanding cuprate superconductivity, as it places strong constraints on any theory. We then use a simple model to explain the data: the superconducting gap is known to be spatially inhomogeneous, which results in a distribution of local transition temperatures and naturally leads to percolation. Percolation, and the scale-free fractal structures that emerge from it, is a well-known and ubiquitous phenomenon: first investigated in the context of polymer growth, it has since been formulated as a mathematical concept and applied to systems as diverse as random resistor networks, organic molecular gels, dilute magnets, the spread of diseases, and the large-scale structure of the universe[27,28]. The basic ingredient in percolation theory is inhomogeneity, and we find that evoking $T_c$ inhomogeneity is essential to understand superconductivity emergence in the cuprates. Remarkably, the minimal percolation model that we employ is sufficient to capture the observed unusual exponential temperature- and magnetic-field dependences of the nonlinear conductivity. We also report complementary linear conductivity measurements and take a fresh look at prior experimental results (torque magnetometry[15], resistivity[7], Seebeck coefficient[8], specific heat[29], and tomographic density of states[19]), to demonstrate that the emergence of superconductivity can be consistently explained with this minimal model. Finally, the universal scale $\Xi_0$ is shown to be a direct measure of the superconducting gap distribution width. The underlying inhomogeneity therefore is unrelated to material details, and must be an intrinsic, generic feature of cuprate superconductors.

## Results

**Nonlinear response**. Nonlinear planar response, for current flow along the $CuO_2$ planes, is measured with a sensitive contact-free method[30] (see Methods). The response can be analyzed by decomposing the signal into harmonics,

$$J = \sigma_1 K + \sigma_2 K^2 + \sigma_3 K^3 + \dots , \qquad (1)$$

where $J$ is the response of the sample to an external field $K$ (electric or magnetic), $\sigma_1$ the linear response tensor, and $\sigma_2$, $\sigma_3$, etc., the correction nonlinear tensors. Here, we discuss the third harmonic $\sigma_3$, the lowest-order conventional correction to the linear response (the second-harmonic $\sigma_2$ can only appear if time reversal or inversion symmetry is broken[31] and is not discussed here). In any alternating-field experiment, magnetic and electric fields are related, and therefore it is arbitrary if one designates the signal at frequency $3\omega$ as proportional to nonlinear conductivity or (complex) susceptibility. Complementary linear conductivity measurements are performed with a microwave cavity perturbation technique (see Methods).

**Temperature dependence**. Measurements of the in-plane linear and nonlinear response were performed for three representative cuprate families: a nearly optimally doped sample of $HgBa_2CuO_{4+\delta}$ (Hg1201), a model cuprate system due to its simple structure, high $T_c$, and minimal point disorder effects[32–35]; an optimally doped $YBa_2Cu_3O_{7-\delta}$ (YBCO) sample with 3% of Cu substituted by Zn (YBCO–Zn), where Zn dramatically affects the superconducting properties[36]; and $La_{2-x}Sr_xCuO_4$ (LSCO), spanning a wide range of doping across the superconducting dome (see Table 1). For all samples, $\sigma_3$ exhibits qualitatively the same temperature dependence (Fig. 1a and Supplementary Figure 1): no signal at high temperatures, a peak at a temperature that we designate as $T_c$, consistent with previous work (see Methods), and a step-like feature below $T_c$. The signal magnitude depends on sample size and shape, and thus is normalized to the peak value.

**Table 1 Normal state and superconducting properties of the investigated samples**

| Sample | doping level $p$ | $T_c$ (K) | $\Xi_0$ (K) |
|---|---|---|---|
| Hg1201 | 0.14 | 94.0 | 25.1 ± 1.4 |
| LSCO-0.08 | 0.08 | 17.3 | 29.9 ± 1.4 |
| LSCO-0.125 | 0.125 | 31.0 | 28.2 ± 0.9 |
| LSCO-0.15 | 0.15 | 37.2 | 28.6 ± 0.2 |
| LSCO-0.19 | 0.19 | 30.1 | 29.3 ± 1.5 |
| YBCO-Zn | 0.15 | 60.2 | 26.1 ± 0.6 |

For LSCO, $p = x$, whereas for Hg1201 and YBCO–Zn, the estimate is based on the findings in refs. 32,36, respectively. As described in the text, $T_c$ corresponds to the peak in the nonlinear conductivity; we estimate the error to be less than 1%. $\Xi_0$ is obtained from nonlinear conductivity using a 3D site-percolation model. The uncertainties are from fits to the percolation calculation

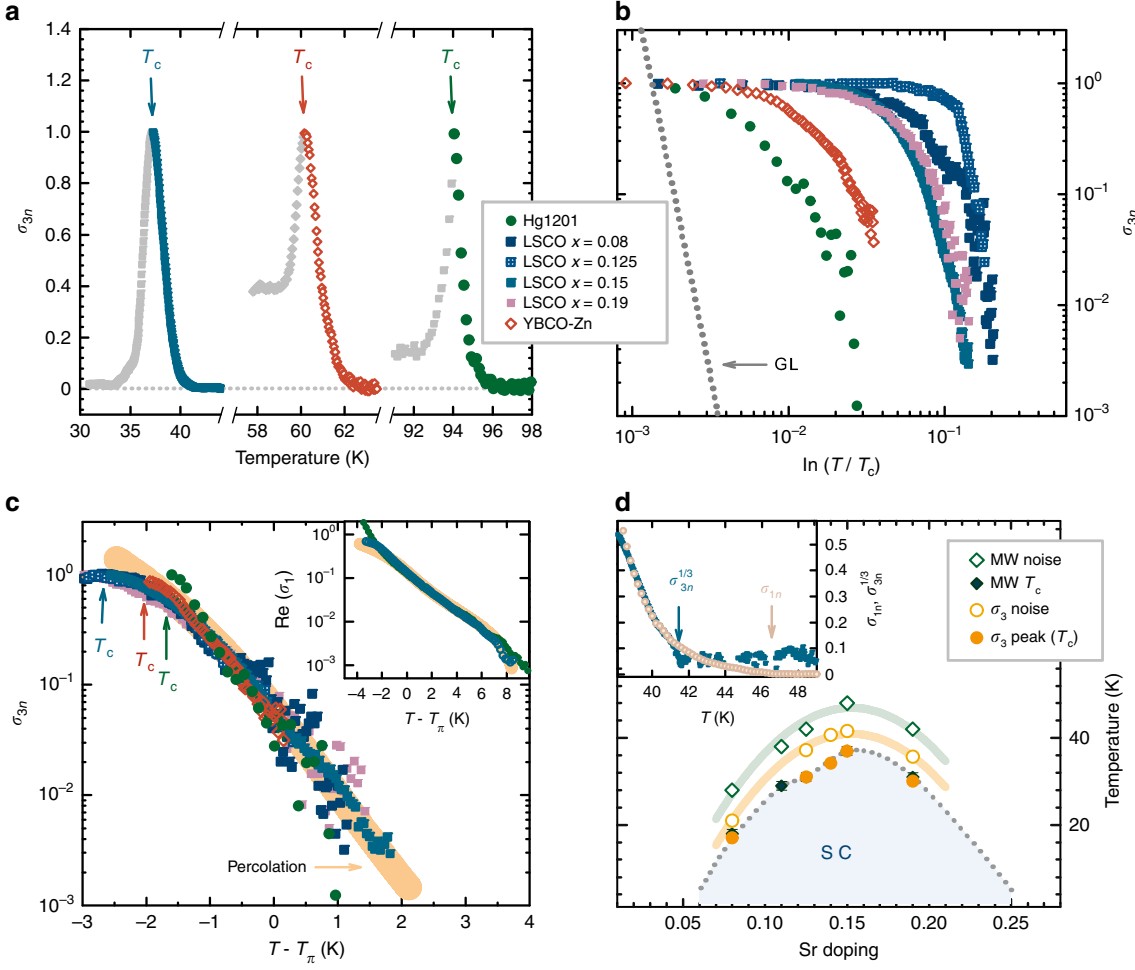

**Fig. 1** Temperature dependence of in-plane linear and nonlinear response in the cuprates. **a** Nonlinear conductivity $\sigma_{3n}$ ($\sigma_3$ normalized to its peak value, which corresponds to the bulk $T_c$ indicated by arrows) for three representative samples close to optimal hole doping: Hg1201, YBCO–Zn, and LSCO-0.15 ($x = 0.15$). **b** Nonlinear conductivity vs. Ginzburg–Landau reduced temperature $\ln(T/T_c)$, which demonstrates that $T_c$ is not the common scale for superconductivity emergence. Dotted line is the GL prediction (see Methods). **c** $\sigma_{3n}$ shifted by a sample-dependent temperature $T_\pi$ collapses to a single curve. This demonstrates the existence of a universal emergence temperature/energy scale. $T_c$ is indicated by arrows (LSCO-0.15, LSCO-0.19, and LSCO-0.08 have indistinguishable $T_c$ values on this scale; $T_c$ for LSCO-0.125 is at $T - T_\pi = -4.5$ K). Orange line is the prediction of the simple site-percolation model discussed in the text. Inset: linear conductivity of Hg1201 and LSCO-0.15 along with the model prediction. **d** Phase diagram of LSCO with the characteristic temperatures below which the superconducting response is first resolved in both linear (microwave—MW) and nonlinear conductivity; errors are determined from the root-mean square noise level and are within the symbol size. These temperatures are significantly different despite the similar signal-to-noise ratio of ~$10^3$ (at $T_c$), consistent with the percolation model. The positions of the peaks in $\sigma_{3n}$ and in the real part of $\sigma_1$ give consistent $T_c$ values. Error bars for $T_c$ are given by the error of the peak temperature determination (and are within the symbol size for $\sigma_{3n}$). Lines are guides to the eye. The inset demonstrates, for LSCO-0.15, that $\sigma_{3n} \propto \sigma_{1n}^3$ until the third-order response is indistinguishable from noise. The two temperatures below which signals are resolved from noise are marked with arrows

We note that $T_c$ as obtained from $\sigma_3$ agrees with the temperature of the peak in the real part of the linear microwave conductivity, which in turn corresponds to the value determined from magnetic susceptibility measurements[12].

The measurements clearly show that the nonlinear response decays quickly above $T_c$, which demonstrates the absence of extended fluctuations. Some previous investigations suggested agreement between experiments and GL theory (with various modifications to the theory[4,7,8,12]) for particular cuprate compounds at particular doping levels; in line with these investigations, we have attempted to analyze our results within the GL framework. Within this framework, we would expect an approximately power-law temperature dependence of $\sigma_3$ (see ref. [37] and Methods), and a scaling of the data for different compounds with the characteristic scale $T_c$. Figure 1b shows our nonlinear conductivity data in dependence on the GL-reduced

temperature $\ln(T/T_c)$ compared to a calculation of $\sigma_3$ using anisotropic GL theory beyond mean field (see Methods), similar to ref. [12]. The theory predicts a temperature dependence of $\sigma_3$ that is clearly incompatible with experiment; the agreement cannot be improved by any tuning of the parameters, such as a different definition of $T_c$ (see Supplementary Figure 2). Even more importantly, the expected scaling is absent: $T_c$ is not the characteristic temperature scale for superconductivity emergence. The scaling argument is valid regardless of the manner in which GL theory is modified. However, the data are remarkably similar on an absolute temperature scale: a simple shift by a sample-dependent temperature $T_\pi$ (that is slightly larger than $T_c$) leads to the data collapse shown in Fig. 1c. This implies that a mechanism that gives rise to approximately exponential behavior with a single temperature scale $\Xi_0$ underlies the emergence of superconductivity. A similar exponential dependence can also be deduced from

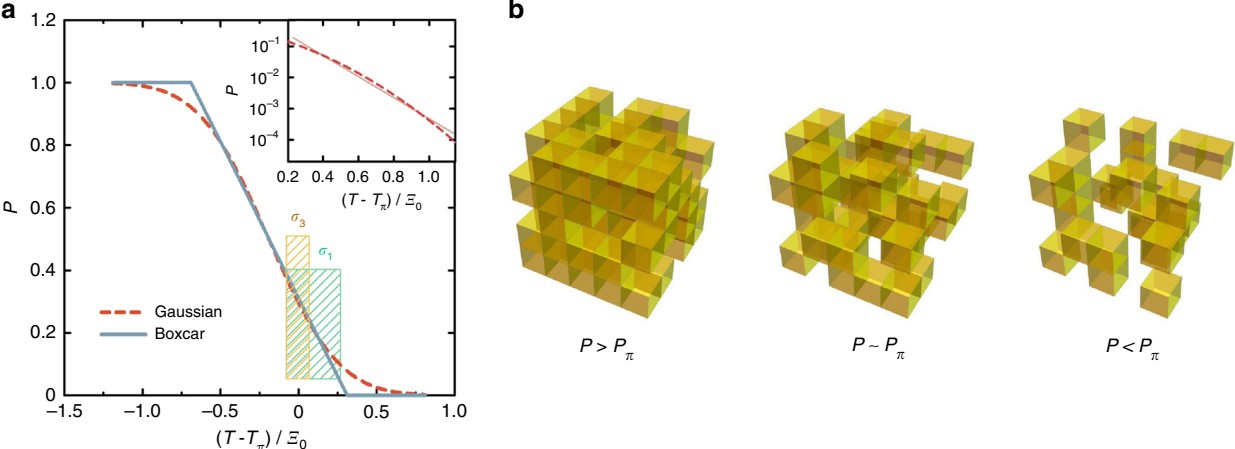

**Fig. 2** The superconducting percolation model. **a** Comparison of two different assumptions for the dependence of the superconducting volume fraction $P$ on temperature, for 3D site percolation. The solid line is the linear assumption—equivalent to a boxcar distribution of local gaps—and the dashed line is obtained if a Gaussian local gap distribution is assumed. The widths of the distributions are taken to be the same and equal to $\Xi_0$. The shaded areas indicate the temperature ranges in which linear and nonlinear conductivity are appreciable in the experiment; this demonstrates that the linear assumption is justified in our case. The inset shows the tail of the integrated Gaussian distribution (dashed line) compared to a pure exponential decay (solid line) on a log-linear scale. The difference is relatively small across four orders of magnitude. Other distributions result in an exact exponential tail, e.g., the gamma and logistic distributions are asymptotically exponential. **b** Schematic representation of the superconducting site-percolation model in a cubic geometry. Yellow patches are superconducting. Above the critical concentration $P_\pi$ (below $T_\pi$), a sample-spanning superconducting cluster exists

linear conductivity (inset in Fig. 1c) and torque magnetometry[15] experiments, indicating its robustness. Clearly a framework other than GL is needed to explain the data.

Since nanoscale electronic inhomogeneity is well documented in the cuprates, e.g., from nuclear magnetic resonance[38–40] and scanning tunneling microscopy (STM)[17,41] measurements, we now attempt to gain understanding through a simple percolation model. The basic ingredient of the model is spatial inhomogeneity of local superconducting gaps, with a distribution width that is characterized by the scale $k_B\Xi_0$. This distribution corresponds to superconducting patches that proliferate upon cooling, and macroscopic superconductivity then emerges via a percolative process[42,43]. We calculate the response assuming nearest-neighbor site percolation, although the result does not critically depend on the details of the scenario (see Methods). For simplicity, we take the material to be made of perfectly connected square or cubic patches that are either nonsuperconducting, with a normal resistance $R_n$, or superconducting, with a nonlinear resistance $R_s(j)$ that depends on the current through the patch $j$ (see Methods). Since we normalize the experimental nonlinear conductivity, we can also normalize the resistances by taking $R_n = 1$. The fraction $P$ of superconducting patches depends on temperature: $P \to 0$ at high temperatures and $P \to 1$ well below $T_c$. At the critical concentration $P_\pi$, the system percolates—a connected, sample-spanning superconducting cluster is formed. $P_\pi$ only depends on the dimensionality of the system[27] and on the details of the percolation scenario (e.g., site vs. bond percolation), and it corresponds to the temperature $T_\pi$ that can be viewed as the "true" underlying resistive $T_c$ in the limit of small currents. In principle, the full temperature dependence of $P$ can be obtained from the underlying gap distribution, but the distribution must be known (or assumed). However, to lowest order, any reasonable distribution yields a linear dependence of $P$ on temperature close to $P_\pi$ (see Fig. 2). We, therefore, approximate $P_\pi - P = (T - T_\pi)/\Xi_0$. The temperature-dependent linear and nonlinear responses are then obtained via an effective medium calculation (see Methods), which yields functions that decay nearly exponentially, in very good agreement with the experimental $\sigma_3$ and $\sigma_1$ (Fig. 1c). Within the nearest-neighbor site-

percolation model, two values of $P_\pi$ are possible: ≈0.3 for three-dimensional (3D) and ≈0.6 for two-dimensional (2D) percolation. Better agreement is obtained with $P_\pi \approx 0.3$ (see Supplementary Figure 3), which suggests essentially 3D superconductivity emergence[27] in the samples studied here. We note that we study the in-plane response, and thus the only role of inter-plane coupling in the percolation model is to determine the effective dimensionality, and hence the percolation threshold.

For $P_\pi \approx 0.3$, the resultant characteristic scale $\Xi_0$ lies in a narrow range for all investigated samples (see Table 1), $\Xi_0 = 27 \pm 2$ K, and hence is de facto universal (the stated uncertainty is 1 s. d. from the mean of the data in Table 1). If we assume 2D percolation and $P_\pi \approx 0.6$, the agreement between $\sigma_3$ and $\sigma_1$ is not as good, and the corresponding $\Xi_0$ is smaller by about a factor of two. We emphasize that the calculated $\sigma_3$ is effectively insensitive to model details such as the parameters of the patch nonlinear response $R_s$, rendering $\Xi_0$ the sole parameter (within a given percolation model). This insensitivity to specifics is a consequence of percolation physics, where model details are unimportant close to the threshold and the response of the largest clusters dominates.

An important feature can be inferred from the comparison of linear and nonlinear response. Within the effective medium calculation, the linear conductivity determines the net current through the sample, given an applied electric field. Yet the nonlinear resistance of the superconducting patches, $R_s$, is current dependent. The third-harmonic signal therefore depends on the third power of the current (to lowest order), which implies that $\sigma_3 \propto \sigma_1^3$. This is indeed borne out by experiment, as seen in Fig. 1d. In contrast, in GL theory both responses are determined by the electric field, and their ratio has a more complex temperature dependence (see ref. 37 and Methods). The *apparent* characteristic temperature scales for $\sigma_3$ and $\sigma_1$, therefore differ because of the nonlinear nature of $\sigma_3$, but the underlying scale $\Xi_0$, which determines the range of superconducting pairing emergence, is the same for both responses. This also implies that the superconducting contribution to the linear response should be discernable up to significantly higher temperatures than the nonlinear part, if the experimental signal-to-noise ratios are

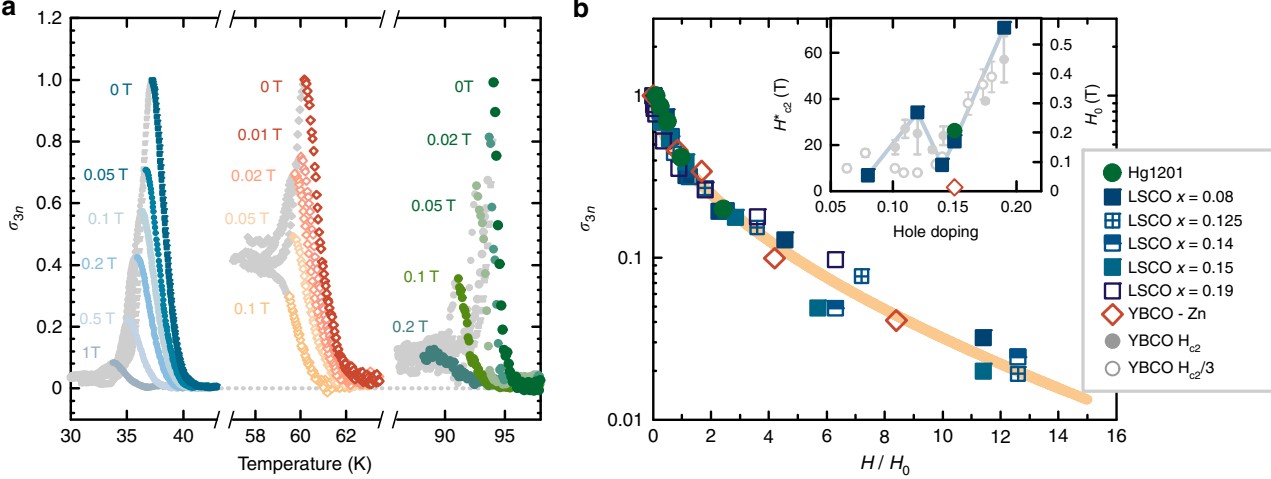

**Fig. 3** Influence of external $c$-axis magnetic field on in-plane nonlinear response. **a** Nonlinear conductivity at various applied fields for Hg1201, YBCO-Zn, and LSCO-0.15, normalized to the zero-field peak values. **b** Scaling of the nonlinear conductivity with magnetic field. Symbols indicate peak values of $\sigma_3$ upon subtracting the high-field step and normalizing to the zero-field values. The orange line is the result of an effective medium calculation for the percolation model (see text). Inset: suppression field $H_0$ compared with previous results for $H_{c2}$ for YBCO obtained using muon spin rotation[44] (full circles) and transport[45] (empty circles). The light blue line is a guide to the eye. Errors (1 s.d.) in $\sigma_{3n}$ and $H_0$ are within the symbol size

similar. Measurements throughout the phase diagram of LSCO consistently confirm this trend (Fig. 1d), which strongly supports the percolation model.

**Magnetic-field effect**. As a further test of the model, we investigate the influence of an external magnetic field on the emergence regime. Although the quantitative effects of a magnetic field are difficult to determine within our simple effective medium approach, we can make qualitative predictions. One would expect the field to greatly influence the superconducting percolation process. Both the critical current of a superconducting patch and the number of patches decrease with increasing field. Above a characteristic field $H_0$, the critical currents of all patches are small, except for the sample-spanning cluster at temperatures below $T_\pi$. At fields significantly above $H_0$, the nonlinear response should therefore only exhibit a step-like feature close to $T_\pi$. Furthermore, $H_0$ is related to the macroscopic critical field $H_{c2}$, as both fields are determined by the underlying superfluid stiffness: $H_0$ describes the properties of the finite-sized clusters below and above $T_\pi$, whereas $H_{c2}$ pertains to the sample-spanning cluster below $T_\pi$. In agreement with these expectations, we find that an external magnetic field strongly suppresses the nonlinear response, rendering it step-shaped far above $H_0$ (Fig. 3a). Once the high-field step-like response is subtracted (see Supplementary Note 4 and Supplementary Figure 5), the data for all samples exhibit universal scaling (Fig. 3b). We apply the same effective medium calculation as for the temperature dependence, assuming a phenomenological power-law dependence of the effective patch critical current on $H/H_0$ (see Supplementary Note 4 and Supplementary Figure 4), and find good agreement with experiment (Fig. 3b). For $H \gg H_0$, only large superconducting clusters survive. Since the cluster-size distribution in any percolation model is generally exponential for the largest clusters[27], this leads to an exponential-like field dependence of the high-field response, as also observed in prior torque measurements[15].

$H_0$ is about two orders of magnitude smaller than $H_{c2}$, consistent with the percolation scenario, since $H_0$ is a property of the *average* (small) cluster. As seen from Fig. 3b, the doping dependencies of the two characteristic fields are remarkably similar, including a minimum close to the "1/8 anomaly" of

LSCO and YBCO[8,44,45]. The substitution of 3% Cu with Zn in YBCO causes a dramatic decrease of $H_0$, in agreement with established effects of Zn on superconductivity in cuprates[36].

## Discussion

Previous reports suggest that percolation processes might play a role in understanding the properties of the cuprates[42] (see also Supplementary Note 3). However, our work demonstrates for the first time that a universal percolation process can describe the prepairing regime. The percolation picture is in excellent agreement with the temperature and magnetic field dependencies of $\sigma_1$ and $\sigma_3$, and one would expect it to provide an explanation of other experimental results as well. Qualitatively, several previous studies indicate that the superconducting precursor appears within a roughly constant temperature range above $T_c$, similar to Fig. 1d; this is visible, e.g., in high-frequency conductivity measurements[6,11,12], specific heat results[46], and resistivity curvature plots[47]. More quantitatively, Fig. 4 demonstrates the similarity of superconducting precursor in several observables. An exponential tail is observed in the dc conductivity[7] of YBCO at various hole doping levels, with a universal slope (Fig. 4b). Notably, YBCO in particular is structurally complex, with alternating $CuO_2$ planes and CuO chains whose filling depends on oxygen concentration; the exponential behaviour, however, is robust and does not depend on the arrangement of the chains. The Nernst effect[8] in Eu-LSCO shows an exponential dependence as well (Fig. 4c). Although this measurement can be described by 2D Gaussian theory close to $T_c$, where corrections to the simple percolation picture are expected, once the data are plotted on an absolute temperature scale, the exponential tail is apparent and reveals the same underlying temperature/energy scale $\Xi_0$. Torque magnetometry measurements on several cuprate families, including underdoped LSCO, bismuth cuprates and Hg1201[15], as well as YBCO[16], exhibit both an exponential signal decrease above $T_c$ (Fig. 4d) and a universal temperature scale $T_d$. The exponential dependences at temperatures well above $T_c$ are a consequence of the tail of the superconducting gap distribution, and for $\sigma_1$ and $\sigma_3$ the effective medium calculation smoothly continues this dependence down to $T_c$. Finally, roughly exponential tails are observed above $T_c$ in specific heat studies[26,46,48]; it is possible to

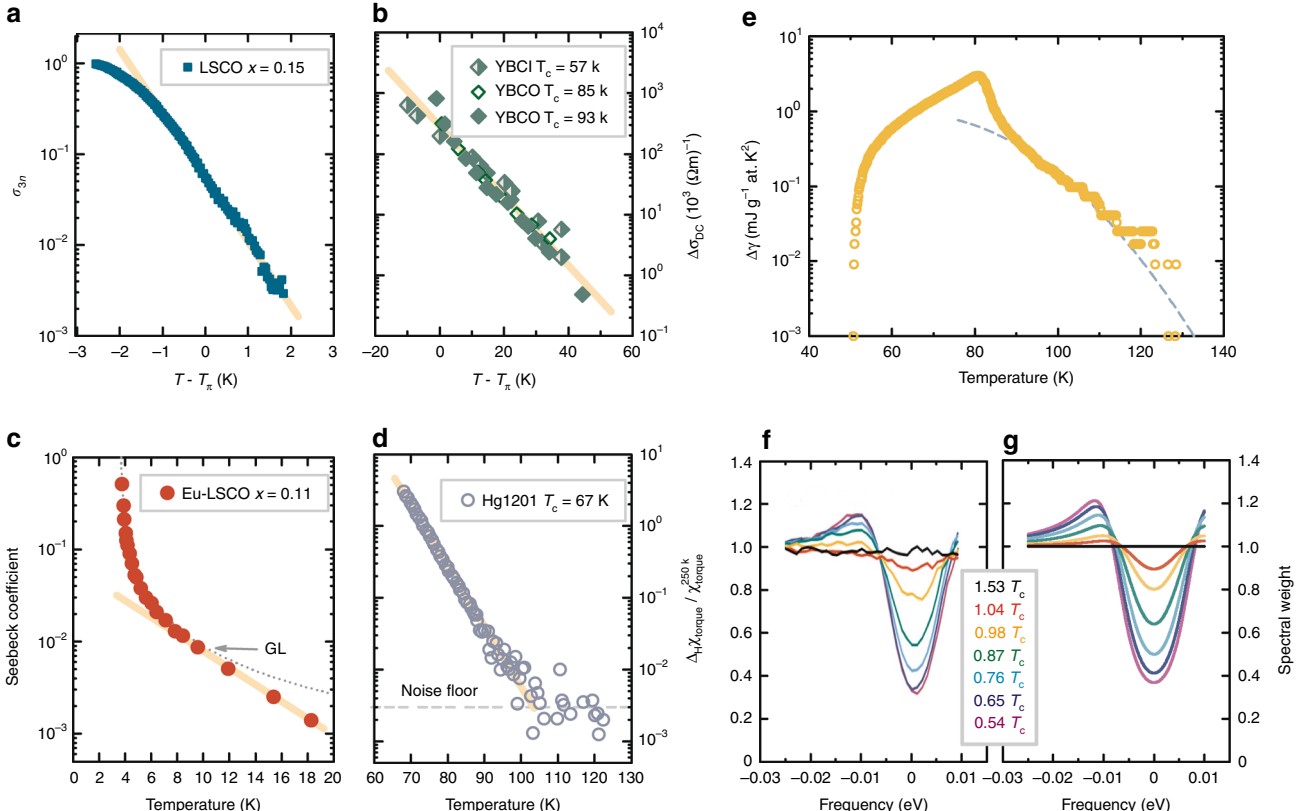

**Fig. 4** Universal percolation physics in the cuprates. **a–d** Four different experimental probes show qualitatively similar non-GL behavior (full lines) above $T_c$: **a** nonlinear conductivity of optimally doped LSCO (this work); **b** superconducting contribution to DC conductivity of YBCO, adapted from ref. [7]—**c** Seebeck coefficient of europium co-doped LSCO ($x = 0.11$) adapted from ref. [8]—the dotted line is the 2D GL prediction valid within 2 K of $T_c$; **d** torque magnetometry in underdoped Hg1201 ($T_c = 67$ K)—the dashed line indicates the noise floor. Adapted with permission from Yu et al.[15]. The decay constants of different observables are proportional to the universal scale $\Xi_0$, but with different prefactors that can be directly calculated from the suggested model (see Methods). **e** Specific heat coefficient. The orange circles are measured values of the superconducting contribution to the specific heat coefficient, $\Delta\gamma$, for $Y_{0.8}Ca_{0.2}Ba_2Cu_3O_{6.75}$ (from ref. [29]), and the dashed line is a calculation that convolutes a mean-field step in $\gamma$ at the local $T_c$ with a Gaussian distribution of $T_c$. The line is obtained with a distribution width of 35 K, similar to the value of $\Xi_0$; it is similar to the dashed line in Fig. 2a, since in a mean-field picture the specific heat essentially measures the superconducting fraction above $T_c$. The measurements also show a fluctuation peak around $T_c$, which is not included in the calculation. **f** Measured tomographic density of states for optimally doped $Bi_2Sr_2CaCu_2O_{8+x}$ reproduced from ref. [19], with permission from APS. **g** Calculated density of states, assuming a Gaussian gap distribution with full width of 3.2 meV (see Methods). In **a–d** the lines are not the result of a direct calculation, but rather highlight pure exponential decay

calculate this within the percolation model by convoluting the mean-field specific heat step at $T_c$ with the gap distribution function (see Methods). This procedure yields agreement with experiment and reveals a scale $\Xi_0$ similar to that obtained from conductivity (Fig. 4e). Notably, critical fluctuations are observed in the specific heat close to the macroscopic $T_c$, which could be also important for other observables in a small temperature range around $T_c$.

We note that the percolation model discussed here is somewhat different from the standard textbook case, in that both the normal and percolating (superconducting) patches have nonzero conductivity. Therefore, instead of a discontinuity at $T_\pi$ and power-law behavior above the percolation temperature (that is predicted if one of the phases is insulating[27]), the calculation yields smooth exponential-like behavior. Yet the underlying distribution of superconducting cluster sizes should still be scale-free (i.e., follow a power law) close to the percolation threshold. A signature of this might be observed with other experimental probes, e.g., recent optical pump-probe experiments[49] uncovered hitherto unexplained power-law superconducting correlations above $T_c$.

We emphasize that the heterogeneity that gives rise to superconducting percolation is qualitatively different from the disorder discussed previously in the context of "dirty" and granular

superconductors[50–53], and from inhomogeneity induced by doping. In alloys[50] and films[51,52], the electronic mean-free path is extremely shortened by scatterers, while in granular materials differing Josephson couplings between granules cause superconducting percolation[50]. Yet here we find that nanoscale *gap* inhomogeneity is crucial: the superconducting gap, and hence the local $T_c$ displays spatial variations and causes the percolation we observe. Related gap disorder (on scales much larger than the superconducting coherence length) has been employed previously in modeling the magnetization of select cuprate and other superconductors[53], but not applied universally or used to calculate transport properties. Inhomogeneity and a residual zero-temperature component of uncondensed carriers has been shown to be essential to understand low-temperature superfluid density and optical response of several cuprates[54]. Spatial gap inhomogeneity also naturally explains the gap filling recently observed in a tomographic density-of-states photoemission experiment[19]. As demonstrated in Fig. 4f, g, a quantitative description of this result can be obtained simply by positing that the measured density of states is an average over spatial regions with inhomogeneous gaps, again with a distribution width of $k_B\Xi_0 \sim 3$ meV, which further supports the percolation scenario (see Methods for details).

Perhaps the most unexpected result of our study, which covers the doping range from the very underdoped ($p = 0.08$) to the overdoped ($p = 0.19$) part of the phase diagram, is the existence of a (nearly) doping- and sample-independent percolation scale $\Xi_0$, which implies a common intrinsic origin of the gap inhomogeneity in all cuprates, irrespective of material details. Doping does not significantly alter this scale, but affects the macroscopic $T_c$ or, equivalently, the critical percolation temperature. Several distinct types of disorder are generally present in the cuprates: the lamellar structure is intrinsically frustrated, which causes structural inhomogeneity; the hole doping process introduces defects into the crystal structure; and doping a strongly correlated electronic system may induce electronic frustration and inhomogeneity. These different kinds of disorder are typically compound and doping dependent[26,55], and various experimental techniques have been used to study them. Residual resistivity, a measure of point disorder, is compound-dependent, and can be very small in cuprates such as Hg1201[33,34]. Furthermore, quantum oscillation experiments point to a high degree of doping (hole concentration) homogeneity in oxygen-doped compounds such as Hg1201[35], thallium cuprates[56], and YBCO[57]. However, this does not preclude nanoscale electronic inhomogeneity unrelated to point disorder. This reasoning is supported by the fact that we find consistent results for distinctly different cuprates[26]: Hg1201, where doping-related point disorder resides relatively far away from the $CuO_2$ planes; LSCO, which exhibits considerable (La/Sr) point disorder in close proximity to the $CuO_2$ planes; and YBCO–Zn, where Zn directly introduces point disorder within the $CuO_2$ plane.

The cuprates also appear to exhibit inherent structural inhomogeneity, an elegant demonstration of which comes from conductivity and hydrostatic relaxation experiments that show stretched exponential behavior characteristic of glassy materials[55]. Moreover, X-ray experiments find complex fractal interstitial-oxygen-dopant structures linked to percolative superconductivity[58]. Local electrostatic disorder has been studied via nuclear quadrupole resonance and revealed that LSCO and Bi-based compounds exhibit higher levels of such disorder[38,59] than oxygen-doped cuprates such as Hg1201 and YBCO, where the dopant atoms reside far from the $CuO_2$ planes[39,40]. Importantly, however, none of these experiments directly detect superconducting gap disorder, making it difficult to establish a relationship between electrostatic/doping inhomogeneities and superconducting gap distributions. STM does probe local gap distributions on the sample surface, but has been applied only to a select number of cuprates, and it is not trivial to separate the superconducting gap from the more inhomogeneous higher-energy (pseudo)gap[17]. We emphasize that the gap distribution (with width $k_B \Xi_0$) relevant for our model likely is not precisely the same as the gap distribution seen by STM, but rather a coarse-grained distribution of mean local gaps (averaged over the local superconducting coherence lengths). Moreover, we expect the distribution to be effectively narrower below $T_c$ because of proximity effects, i.e., large-gap superconducting regions may induce a gap in neighboring regions. Nevertheless, STM clearly reveals disorder structures in both underdoped and overdoped[41] $Bi_2Sr_2CaCu_2O_{8+\delta}$, and extended analysis shows a correlation between the presence of inhomogeneous high-energy gaps and superconductivity[60]. A recent phenomenological model[61] based on inhomogeneous, temperature and doping dependent (de)localization of one hole per primitive cell can explain the main features of the cuprate phase diagram and superconductivity. It is conceivable that a universal scale $k_B \Xi_0$ emerges via a complex renormalization of these high-energy localization gaps[61] (see also Supplementary Note 3). In this case, the gap disorder would not necessarily be related to any local doping inhomogeneity: the material may be homogeneously doped, yet possess an underlying gap distribution.

The existence of exponential behavior with a universal scale $\Xi_0$ also shows that expected GL superconducting fluctuations are considerably weaker than inhomogeneity effects. Conversely, if GL fluctuations were important, the simple percolation model with a single $\Xi_0$ would not describe the measurements. This furthermore points to 3D percolation—except perhaps in the special case of strong stripe correlations in La-based cuprates[8,62] such as $La_{1.875}Ba_{0.125}CuO_4$ and $La_{1.8-x}Eu_{0.2}Sr_xCuO_4$ (see Supplementary Note 2)—since the strong vortex–antivortex fluctuations expected in the 2D case should significantly broaden the onset of superconductivity.

To conclude, we have employed a new approach to the cuprate prepairing problem by studying nonlinear conductivity. The unexpected scaling of nonlinear and linear conductivity for widely different cuprates constitutes a benchmark result for any theory of superconducting prepairing in these materials. Taking into account the well-established fact that significant gap inhomogeneity is present in the cuprates, we have provided a simple framework in which inhomogeneity plays a pivotal role. Our results thus show that intrinsic and universal superconducting gap inhomogeneity is highly relevant to understanding the superconducting properties of the cuprates.

## Methods

**Samples**. The Hg1201 and LSCO samples are single crystals of well-established high quality used in previous work[63,64] with volumes of about 1 mm³. Hg1201 is grown using an encapsulation method, while LSCO crystals are grown in a traveling floating zone furnace. YBCO–Zn is an oriented powder sample with 3% Cu substituted with Zn, which enables us to discern effects of intentionally introduced $CuO_2$ plane disorder. This sample was prepared using a standard solid-state reaction, used in prior Zn nuclear quadrupole resonance experiments and characterized in detail[36]. See Table 1 for additional sample information.

**Linear and nonlinear conductivity**. Nonlinear response measurements typically require relatively large applied fields in order to detect the small signals. Hence, the most serious problem that plagues nonlinear measurements of conductive systems is Joule heating, i.e., the variation of the conductivity/susceptibility with temperature induced by resistive heating of the sample. If a constant or slowly varying electric field is used to detect nonlinear response, the large current will heat the sample during measurement, and spurious nonlinear contributions will appear if the resistance depends on temperature. Conventionally, millisecond field pulses are used to alleviate the heating problem, but heating still plays a role for highly conducting samples and needs to be disentangled from other possible contributions[65,66]. A pivotal step in our experiment is the use of a high-frequency excitation field—if the frequency is high enough, the time-dependent temperature change of bulk samples cannot follow the rapidly changing field, and no heating-induced nonlinear signal is observed. An average, time-independent heating is still present, but this does not influence the measurement of the nonlinear response. In principle, time-independent heating may cause a small shift of the sample temperature, but this was determined to be negligible in our case from a comparison of $T_c$ (peak positions) in linear and nonlinear conductivity throughout the phase diagram of LSCO. The nonlinear conductivity experiments are performed with a contactless radio-frequency two-coil setup, with excitation frequency $\omega/2\pi = 17$ MHz and phase sensitive detection at $3\omega/2\pi$ using a Stanford Research Systems SR844 RF lock-in amplifier. The coil system is kept at the constant temperature of the liquid helium bath, while the sample temperature is varied independently. We use a non-resonant circuit (a coil with silver paint serving as a distributed capacitance) for excitation, and a tuned resonant LC circuit for detection. A thin-walled glass tube separates the vacuum of the sample space from the liquid helium bath and introduces no distortions to the signal. The sample is mounted on a sapphire holder with temperature control sensitivity better than 1 mK. The setup was previously tested under various conditions[30,67]. Notably, a similar methodology was used in the past to study the nonlinear Meissner effect at low temperatures[68]. The electric fields with the samples may be estimated using Maxwell's equations, which gives an amplitude $E \sim B\omega L$, where $B$ is the magnetic field amplitude, $\omega \sim 2\pi \cdot 20$ MHz the oscillation frequency, and $L \sim 2$ mm the typical linear sample dimension. The amplitude of the magnetic field was deliberately kept small, and estimated to be about 0.1 G from the characteristics of the excitation circuit and coil. The electric field amplitude is then $E \sim 0.02$ V/cm.

We performed complementary microwave (linear) conductivity experiments with a resonant cavity perturbation technique[69] extensively used to study cuprate superconductors[5,12]. The sample was mounted in an evacuated elliptical microwave

cavity made of copper and immersed in a liquid helium bath. The complex conductivity of the sample was obtained by measuring the temperature dependence of the Q-factor and resonant frequency of the cavity by recording the cavity resonance curve using microwave frequency modulation and employing a demodulator. The signal from the demodulator was fed into a SR830 lock-in amplifier and the Q-factor determined from measurements of the higher harmonic components of the modulation frequency. The microwave cavity resonance was close to 10 GHz, while the modulation frequency was 990 Hz. Similar to the nonlinear conductivity experiment, the cavity was kept at constant temperature, while the sample temperature was varied in a wide range. We obtained the superconducting response above $T_c$ by subtracting the conductivity measured with an external magnetic field of 16 T (perpendicular to the CuO$_2$ planes) from the zero-field conductivity. No appreciable difference in conductivity was observed between 12 and 16 T in the relevant temperature range.

**GL theory.** Classic GL superconducting fluctuations have been extensively investigated in the cuprates using linear response in a wide frequency range[3–7,11,12,70–72]. Nonlinear response is a better probe of fluctuation contributions, since in linear response one must always attempt to determine and subtract a normal-state contribution, a complication that is absent in the third harmonic. At a quantitative level, the linear and nonlinear GL-fluctuation response has been calculated beyond mean field[37], and with included anisotropy[71–73] (only linear conductivity). For an isotropic type-II superconductor, the nonlinear conductivity in both the Gaussian and critical fluctuation regimes is shown to be proportional to the linear conductivity, as follows. In general, one can define a field-dependent conductivity, $\sigma(E) = \sigma(E = 0)\Sigma(E)$. The scaling function $\Sigma(E)$ has different forms in different fluctuation regimes and for small/large electric fields, but in the small-field approximation the leading term is always $1 + A(E/E_0)^2$, where $A$ is a numerical constant and $E_0$ a reference electric field[37]. The field $E_0$ depends on temperature through the mean-field correlation length ($\xi$), as $E_0 \sim \xi^{-3}$. Therefore, $\sigma_3 = \sigma_1 A/E_0^2 \sim \sigma_1 \xi^6$. Since, in GL theory, the linear and nonlinear responses are due to the same fluctuation physics, such a scaling relationship between the two should hold regardless whether $ab$-plane/$c$-axis anisotropy and short-wavelength cutoffs[70–73] are included. Thus one can directly compare the temperature dependence of the linear/nonlinear response to the predictions of linear GL-fluctuation theory. Due to the low frequency of our experiment, the linear response simply corresponds to the in-plane dc linear conductivity, $\sigma_{ab}^{DC}$. The dc conductivity is given by[71]

$$\sigma_{ab}^{DC} = \frac{e^2}{16\pi\hbar\xi_{0c}}\left(\frac{\xi_{ab}(T)}{\xi_{0ab}}\right)^{z-1} f(Q_{ab}, Q_c), \qquad (2)$$

where $\xi$ is the superconducting coherence length, the indices $ab$ and $c$ correspond to in-plane and $c$-axis quantities, respectively, $z$ is the dynamical exponent, and $f(Q_{ab}, Q_c)$ a function of the temperature-dependent anisotropic fluctuation cutoffs $Q_{ab}$ and $Q_c$ in reciprocal space. The cutoffs are $Q_{ab,c} = \sqrt{3}\Lambda_{ab,c}\xi_{ab,c}(T)/\xi_{0ab,c}$, where $\Lambda_{ab,c}$ are temperature-independent cutoff scales. Since the electric fields applied in our measurements are small, they are significantly below $E_0$ (except perhaps in the closest vicinity of $T_c$, where $E_0$ rapidly goes to zero).

The relevant dimensionless temperature variable for $\xi(T)$ in GL theory is $\ln(T/T_c)$, and data for several cuprates are plotted vs. this GL reduced temperature in Figs. 1b and S2. The theoretical prediction obtained from Eq. (1) using the realistic parameters[12] $\Lambda_{ab} = 0.1$ and $\Lambda_c = 0.02$ is shown in Fig. 1b. The theoretical prediction clearly decays much faster than the data for all investigated samples. We note that the choice of a different value for $T_c$ cannot improve the agreement between data and theory. This is demonstrated in Supplementary Figure 2 for the case of LSCO with $x = 0.15$ (measured $T_c = 37.2$ K). Better agreement can be obtained if the reduced temperature variable is multiplied by material-dependent constants for different samples, but in the case of GL fluctuations these would be additional arbitrary nonuniversal free parameters without obvious physical meaning. Even more importantly, the shape of the temperature dependence cannot be satisfactorily reproduced by GL theory, whatever scaling one employs on the temperature axis.

**The minimal percolation model.** The main idea of the model is that nanoscale superconducting patches form and proliferate in the material (Fig. 2b), and that macroscopic superconductivity then emerges via a percolation process. We assume perfectly connected square or cubic patches (2D or 3D nearest-neighbor site percolation) that are either nonsuperconducting, each with a normal resistance $R_n$, or superconducting, each with a nonlinear resistance[43]

$$R_s(j) = R_0 + \frac{R_n - R_0}{e^{-4(j-J_c)} + 1}, \qquad (3)$$

where $j$ is the current through the superconducting patch, $R_0$ its residual resistivity (due to the finite size of the patch, and $R_0 << R_n$), and $J_c$ the patch critical current; $j$ and $J_c$ are dimensionless currents. We assume the patches to be static (which is probably not a good approximation well above $T_c$) and neglect Josephson couplings and proximity effects (not a good approximation very close to $T_c$). The fraction of

superconducting patches is taken to be $P$, with $P \to 0$ at high temperatures and $P \to 1$ well below $T_c$. The critical concentration at which the system percolates, $P_\pi$, depends on the dimensionality of the system and the chosen percolation model. In the nearest-neighbor site-percolation scenario used here, we have[27] $P_\pi \approx 0.3$ (3D) and $P_\pi \approx 0.6$ (2D). Site percolation is physically realistic in the case of superconducting patches with different local $T_c$ values, as seen in STM[17], but the particular choice of the percolation model does not critically affect the modeling, as we show below. In order to make a quantitative comparison to experiment, a dependence of $P$ on temperature must be assumed. The simplest possibility is a linear dependence, $P_\pi - P = (T - T_\pi)/\Xi_0$, where $\Xi_0$ is the universal temperature/energy scale that connects $P$ and $T$. Physically, the linear dependence is equivalent to taking the distribution of local superconducting gaps to be a simple boxcar function of width $\Xi_0$. Yet the linear term is the leading term for any realistic distribution, and thus this approximation is always valid not too far from $T_c$. Our goal, in the spirit of the minimal model, is to avoid any assumptions related to the gap distribution. This approach in the case of linear and nonlinear conductivity gives good results. We illustrate the difference between our assumption of a linear dependence of $P$ on $T$ and a more realistic Gaussian distribution of local gaps in Fig. 2a. At high temperatures, the Gaussian distribution results in better asymptotic behavior, which eliminates the artificial cutoff present in the linear approximation and gives rise to exponential tails of conductivity, magnetization, etc. Yet in the temperature range where linear and nonlinear conductivity is measurable, the differences are minimal.

The linear and nonlinear responses are calculated via effective medium theory[74], using the form most appropriate for site percolation[75]. Since the experimental nonlinear response is normalized, we also normalize the calculated response by $R_n$ (i.e., take that $R_n = 1$). In order to calculate the third-order nonlinear conductivity, the dependence of the voltage on current was determined, and $\sigma_3$ was obtained through an expansion in powers of voltage. Due to the percolative nature of the system, $\sigma_3$ is insensitive to the values of $R_0$ and $J_c$ in the region of interest close to $T_\pi$ (as long as the current $j$ is much smaller than $J_c$). Thus the only parameters entering the calculation of $\sigma_3$ are $\Xi_0$ and the percolation threshold concentration $P_\pi$ of superconducting patches (which depends on the number of spatial dimensions, on site vs. bond percolation, etc.). $R_0$ is used in the linear response calculation, and was determined to be $0.005R_n$, which is realistic for nanoscale patches at a finite excitation frequency[43,50].

In order to obtain $\Xi_0$ and to determine if 3D (with $P_\pi \approx 0.3$) or 2D (with $P_\pi \approx 0.6$) site percolation is more appropriate, we simultaneously calculate the linear and nonlinear conductivity and compare to the measurements (Fig. 1c). Although the results do not critically depend on $P_\pi$, a 3D site-percolation model with $P_\pi = 0.31$ yields the best agreement with the data. For example, it enables the linear and nonlinear response in LSCO to be described with a single $\Xi_0 = 28.0 \pm 0.4$ K, whereas in the 2D model the discrepancy between $\Xi_0$ obtained from linear and nonlinear conductivities differs at least by 25% (Figure S3). With $P_\pi = 0.31$ fixed, individual fits to only the nonlinear response of all investigated compounds (Table 1) gives the overall estimate $\Xi_0 = 27 \pm 2$ K, whereas the simultaneous calculation of both $\sigma_1$ and $\sigma_3$ for LSCO gives the higher precision above. We emphasize that the parameter $R_0$ does not influence the determination of $\Xi_0$: $R_0$ influences the shape of the linear conductivity curve, whereas $\Xi_0$ sets the range of the superconducting contribution. $T_\pi$ is calculated separately in a model-free way to obtain the best data scaling, with typical uncertainties smaller than 0.05 K. The LSCO-0.15 data are taken as a reference since they exhibit the best signal-to-noise ratio. Notably, the determination of $\Xi_0$ is independent of $T_\pi$, since the exponential decay rate constant of $\sigma_3$ is simply inversely proportional to $\Xi_0$ (i.e., the rate is 42.6 K/$\Xi_0$). The calculated curves depart from measurements close to the macroscopic $T_c$, which is expected—once a significant volume fraction of the sample is superconducting, Josephson couplings can no longer be neglected, macroscopic phase coherence sets in, and the simple percolation picture needs corrections.

One can perform a similar effective medium calculation for 3D nearest-neighbor bond ($P_\pi = 0.25$) rather than site percolation, or for any other percolation model with a similar critical concentration, and fit to the nonlinear data. This in itself poses no problems and will increase $\Xi_0$ (by about 20%). However, the linear conductivity provides a constraint—similar to the 2D case, it cannot be simultaneously obtained with the same $\Xi_0$ (the difference being about 10%, larger than the uncertainties). Also, the corrections due to $P$ vs. $T$ nonlinearity may become important. In any case, the difference between $P_\pi = 0.31$ and 0.25 is not very significant in view of the crudeness of the modeling, but the data do support 3D percolation. The cuprate superconductors are known to be strongly anisotropic; in the site-percolation model, this translates to anisotropy within the patches (i.e., they are elongated in the $c$-direction), but this does not change the percolation threshold. Since we measure in-plane response, the threshold is the only important parameter. A possible exception would be systems with effectively decoupled layers (such as Eu–LSCO and LBCO close to doping 1/8, as discussed).

**Modeling of specific heat.** Specific heat measurements in several cuprates[29,48] show high-temperature tails above $T_c$. Here, we show that the tails can be modeled in a quantitative fashion by simply convoluting the standard mean-field step in specific heat at the (local) $T_c$ with the gap distribution. We model the mean-field superconducting contribution to the specific heat coefficient by a simple linear dependence below the local $T_c$, $\Delta\gamma_{loc} = a(T/T_c - 1/2)$, and take it to be zero above

$T_c$. Such a form is not correct at low temperatures, but is appropriate close to $T_c$. The coefficient in a system with a distribution of $T_c$ is then just

$$\Delta\gamma(T) = \int g(T_c)\Delta\gamma_{loc}(T, T_c)dT_c, \qquad (4)$$

where $g$ is the distribution function. The calculated $\Delta\gamma$ is shown in Fig. 4e in comparison with data on $Y_{0.8}Ca_{0.2}Ba_2Cu_3O_{6.75}$ from ref. 29, with a Gaussian distribution of gaps centered at 75 K (the macroscopic $T_c$ is some 6 K larger, in accordance with Fig. 3a). The best agreement with the data above $T_c$ is obtained with a distribution width of 35 K, slightly larger than the values of $\Xi_0$ obtained from linear and nonlinear conductivity in the main text. Close to the macroscopic $T_c$, the measurements also show a fluctuation-induced peak, which is not included in the simple mean-field summation that we have performed. Despite the simplicity, our approach is in acceptable quantitative agreement with the tail in the specific heat coefficient, which is an important confirmation of our model using a bulk thermodynamic probe.

**Percolation interpretation of the tomographic density of states**. Along with transport properties, the percolation model can be used to explain other seemingly unconventional results for the cuprates. One example is the tomographic density of states (TDOS) obtained in recent photoemission measurements[19]. The effective gap obtained in these experiments does not close at the macroscopic $T_c$, but a "filling" of the density of states is observed to extend to temperatures ~$1.2T_c$ (Fig. 3f). The gap filling was attributed to an increased superconducting pair-breaking rate, and the response above $T_c$ to preformed pairs. However, as we now show, both effects arise naturally if one assumes a spatial gap distribution. In ref. 19, the density of states was fitted to the standard expression

$$\rho_{Dynes} = \mathrm{Re}\frac{\omega - i\Gamma}{\sqrt{(\omega - i\Gamma)^2 - \Delta^2}}, \qquad (5)$$

where $\omega$ is the frequency relative to the Fermi level, $\Gamma$ the pair-breaking rate, and $\Delta$ the superconducting gap. To describe the data with this formula, the pair-breaking rate must increase to $\Gamma \sim \Delta$ close to $T_c$, which signals that the description is no longer physically valid. We find that the experimental result can be quantitatively reproduced by employing a temperature-independent $\Gamma$ and by considering that the experiment measures the average density of states in a system with a real-space gap distribution. We then simply convolute the density of states with a gap distribution function, and employ the standard BCS temperature dependence for the gaps. Notably, a similar procedure was recently used to model ARPES data[76] in $Bi_2Sr_2CaCu_2O_{8+y}$. A Gaussian gap distribution with mean $\Delta_m = 9.6$ meV and full width at half maximum $\Delta_0 = 3.2$ meV (in line with ref. 17 and with our nonlinear conductivity measurements) yields rather good agreement with the TDOS experiment at all temperatures (Fig. 3g). This constitutes a strong, independent confirmation of the percolation/gap disorder scenario.

## Data availability

All data needed to evaluate the conclusions in the paper are present in the paper and/or Supplementary information. Additional data related to this paper may be requested from the authors.

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

## Acknowledgments

We thank A.V. Chubukov and J.M. Tranquada for comments on the manuscript. D.P., M.V., M.S.G., and M.P. acknowledge funding by the Croatian Science Foundation under grant no. IP-11-2013-2729. The work at the University of Minnesota was funded by the Department of Energy through the University of Minnesota Center for Quantum Materials under DE-SC-0016371. The work at the TU Wien was supported by FWF project P27980-N36 and the European Research Council (ERC Consolidator Grant no. 725521). We acknowledge M.K. Chan for contributing to Hg1201 sample preparation and characterization.

## Author contributions

D.P. and M.V. built the nonlinear conductivity setup, performed measurements, and analyzed the data. M.S.G. and M.P. built the microwave conductivity setup and performed linear conductivity measurements. M.P. supervised all conductivity experiments. D.P., G.Y., M.G., and N.B. initiated the paraconductivity studies. D.P., M.P. and N.B. conceived the idea to pursue the percolation-based data analysis. D.P. performed the percolation calculations. T.S. prepared the LSCO samples. D.P. prepared the YBCO-Zn sample. G.Y. and N.B. prepared the Hg1201 sample. D.P., M.G., and N.B. wrote the paper with input from all authors.

## Additional information

**Competing interests:** The authors declare no competing interests.

