## [Peer Review file · Nature Communications]

Reviewers' comments:

Reviewer #1 (Remarks to the Author):

- Key results: Please summarise what you consider to be the outstanding features of the work.

Using nonlinear conductivity Pelc et al. detect traces of superconductivity in a relatively narrow range above the nominal superconducting transition temperature in three varieties of cuprate superconductors. Their results are interpreted with a percolation model where the inhomogeneous superconducting gap has a distribution width characterised unexpectedly by a common energy scale.

- Validity: Does the manuscript have flaws which should prohibit its publication? If so, please provide details.

Based on the information provided in the manuscript and supplementary information I did not identify any flaws in the manuscript.

- Originality and significance: If the conclusions are not original, please provide relevant references. On a more subjective note, do you feel that the results presented are of immediate interest to many people in your own discipline, and/or to people from several disciplines?

The paper adds to an existing body of evidence that fluctuating or precursor superconductivity is confined to a more limited temperature envelope than the much larger pseudogap temperature scale. The fluctuation region measured by this technique is on the smaller end of the scale compared to other estimates from specific heat (Tallon et al., PRB 83, 092502 [2011]) or infrared spectroscopy (ref. 11). But is similar to the range quoted by Kokanovic et al. (PRB 88, 060505(R) [2013]) from torque magnetometry.

Due to the apparent failure of Ginzburg-Landau theory to explain the results, the data is interpreted with a percolation model. Percolation models for cuprates have long been suggested, see for example J.C. Phillips PRB 41, 850 (1990). The originality in the present work is the identification of the common energy scale and the use of the nonlinear conductivity method.

The work will be of interest to researchers in the field of high-temperature superconductivity, as well as practitioners of the nonlinear conductivity technique.

- Data & methodology: Please comment on the validity of the approach, quality of the data and quality of presentation. Please note that we expect our reviewers to review all data, including any extended data and supplementary information. Is the reporting of data and methodology sufficiently detailed and transparent to enable reproducing the results?

The data appears to be of high quality and is presented clearly. I suggest a different colour or symbol for the LSCO $x=0.19$ data in fig 1b because it looks too similar to the grey squares in fig 1a. Some of the technical parameters of the model, such as the number of sites, would be useful to list to help readers reproduce the calculations.

- Appropriate use of statistics and treatment of uncertainties: All error bars should be defined in the corresponding figure legends; please comment if that's not the case. Please include in your report a specific comment on the appropriateness of any statistical tests, and the accuracy of the description of

any error bars and probability values.

Uncertainties are reported and the sensitivity of the model calculations to different inputs is discussed.

- Conclusions: Do you find that the conclusions and data interpretation are robust, valid and reliable?

The data interpretation seems to be reliable. The disagreement with Ginzburg-Landau theory is evident.

- Suggested improvements: Please list additional experiments or data that could help strengthening the work in a revision.

The authors could comment on the role of critical fluctuations which are expected to be large in cuprates.

- References: Does this manuscript reference previous literature appropriately? If not, what references should be included or excluded?

Further evidence of pairing above T_c includes:

Photoemission work of Kondo et al., Nature Commun. 6, 7699 (2015).

Specific heat, Tallon et al., PRB 83, 092502 [2011].

Torque magnetometry, Kokanovic et al. PRB 88, 060505(R) [2013]

- Clarity and context: Is the abstract clear, accessible? Are abstract, introduction and conclusions appropriate?

In the abstract I feel it is more appropriate to say that the emergence of superconductivity in the cuprates is "limited" by their intrinsic inhomogeneity, rather than "dominated" by it. Presumably the pairing itself still originates from interactions with some unidentified boson.

Reviewer #2 (Remarks to the Author):

Report of Referee

Nature Communications manuscript NCOMMS-18-09803

Remarks to the Author:

1. Summary of the key results

In this work, Damjan Pelc & co-workers reported experimental measurements of the nonlinear conductivity. The authors claimed the nonlinear conductivity vanishes exponentially above T_c , both with temperature and magnetic field, and exhibits temperature-scaling characterized by a nearly universal scale Ξ_0 , and that their findings are well captured by a simple percolation. The authors

claimed that they have resolved a long-standing conundrum by showing that the emergence of superconductivity in the cuprates is dominated by their intrinsic inhomogeneity.

2. Originality and interest: if not novel, please give references

The phenomena associated with the pairing mechanism responsible for such unexpectedly high T_c in the HTSC have been subject to intense interest for many years, as they are important in terms of interesting, complex physics involved. The most important questions are: what causes superconductivity in the HTSC and crucially why are the transition temperatures so high? If these questions could be comprehensively answered then the pathway to develop new materials with even higher transition temperatures and other improved properties, such as higher critical current densities, would be clarified.

The main problem here is that all the demonstration rests on two unique samples for two representative cuprate families; a nearly optimally-doped sample of $\text{HgBa}_2\text{CuO}_{4+\delta}$ (Hg1201), an optimally-doped $\text{YBa}_2\text{Cu}_3\text{O}_{7-\delta}$ (YBCO) sample with 3% of Cu substituted by Zn (YBCO-Zn) and $\text{La}_{2-x}\text{Sr}_x\text{CuO}_4$ (LSCO) samples, (spanning a wide range of doping across the superconducting dome) are known as samples which exhibits considerable (La/Sr) point disorder.

YBCO has CuO chains. This makes YBCO different from other cuprates. The authors should consider more quantitatively the contributions of CuO chains on their experimental data. It is known that high quality YBCO samples (with sharp superconducting transition temperatures) have several different Cu-O chain superstructures: ortho II (O-II), O-V, O-VIII, O-III. It would be interesting to look the results for these samples.

The open issues and novelty of the paper are not clearly presented. In the introduction, the authors mention several open questions and challenges concerning the subject of superconducting fluctuations in cuprates but the focus of the paper narrows down to the controversial issue of the validity of the Gaussian fluctuation picture and that their findings are well captured by a simple percolation.

The authors should more carefully select the references to previous studies. Their work follows several recent papers on the above controversy and many others that have not been cited. In fact, the controversy is old and experimental evidence of Gaussian fluctuations above T_c was reported by several groups [1-3].

[1] P.P. Freitas, C.C. Tsuei and T.S. Plaskett, Phys. Rev. B 36, 833 (1987).

[2] R. Hopfengartner, B. Hensel, and G. Saemann-Ischenko, Phys. Rev. B 44, 741 (1991).

[3] A. Gauzzi and D. Pavuna, Phys. Rev. B 51, 15420 (1995).

The contribution of this paper for high- T_c superconductivity is not clear. From this result, what can the authors conclude? Is it the way to understand the origin of the pairing mechanism in the HTSC and to increase T_c ?

The experimental data are clear and the authors' analysis of their data may be appropriate. However, the results and discussion are highly specialized and are not suitable for general readers.

In conclusion, I don't think this work has enough novelty, contains new physics and is of significant importance to be published in Nature Communications.

3. Data & methodology: validity of approach, quality of data, quality of presentation

The manuscript is well written with clear explanation of the experimental techniques used. The

presentation of the results is also clear.

4. Appropriate use of statistics and treatment of uncertainties

Good.

Reviewer #3 (Remarks to the Author):

The present manuscript describes microwave resistance measurements on several hole-doped cuprates that claim to show a narrow region in temperature above the superconducting critical temperature in which superconducting fluctuations are present. The subsequent analysis shows that these fluctuations do not conform to the standard Ginzburg-Landau theory for critical fluctuations. On the basis of this analysis, the authors propose a percolation model for the development of superconductivity with decreasing temperature and introduce a new energy scale that is claimed to be universal and related to the distribution of (superconducting) gap magnitudes.

The debate regarding the nature of fluctuations in hole-doped cuprates has raged for a number of years, and in my opinion, this manuscript is a worthwhile contribution to this debate as it provides a different viewpoint that may help to guide future thinking. I am therefore minded to recommend its publication, provided that the referees address the following points:

(I) The data are not presented in a particularly transparent way. I could not deduce, for example, how the σ_3 data were extracted from the raw data. I recommend that the authors provide a couple of figures in the Supplementary Material explaining how the σ_3 plots shown in Figure 1 were obtained.

(II) There is no plot of $P(T)$, i.e. the temperature dependence of the fraction of superconducting clusters. This is an important element in the overall story and its omission needs to be rectified.

(III) Once the $P(T)$ plot is introduced, the authors should discuss how the evolution of $P(T)$ agrees (or not) with the typical behavior of the specific heat jump at T_c . Resistivity is of course a one-dimensional probe of superconductivity, while specific heat is a three-dimensional probe, and its temperature dependence through T_c should reflect to some degree the volume fraction of superconductor within the bulk of the samples.

(IV) The discussion regarding the gap inhomogeneity deduced from STM is confined to Bi2212, which is not studied in the present manuscript. It is well known, e.g. from NMR linewidths, that this is the most inhomogeneous of the hole-doped cuprates. I would recommend the authors to add some qualitative discussion about the variation in inhomogeneity across the various families in the revised manuscript.

Reviewer 1

Using nonlinear conductivity Pelc et al. detect traces of superconductivity in a relatively narrow range above the nominal superconducting transition temperature in three varieties of cuprate superconductors. Their results are interpreted with a percolation model where the inhomogeneous superconducting gap has a distribution width characterised unexpectedly by a common energy scale.

- **Validity:** Does the manuscript have flaws which should prohibit its publication? If so, please provide details.

Based on the information provided in the manuscript and supplementary information I did not identify any flaws in the manuscript.

- **Originality and significance:** If the conclusions are not original, please provide relevant references. On a more subjective note, do you feel that the results presented are of immediate interest to many people in your own discipline, and/or to people from several disciplines?

The paper adds to an existing body of evidence that fluctuating or precursor superconductivity is confined to a more limited temperature envelope than the much larger pseudogap temperature scale. The fluctuation region measured by this technique is on the smaller end of the scale compared to other estimates from specific heat (Tallon et al., PRB 83, 092502 [2011]) or infrared spectroscopy (ref. 11). But is similar to the range quoted by Kokanovic et al. (PRB 88, 060505(R) [2013]) from torque magnetometry.

Due to the apparent failure of Ginzburg-Landau theory to explain the results, the data is interpreted with a percolation model. Percolation models for cuprates have long been suggested, see for example J.C. Phillips PRB 41, 850 (1990). The originality in the present work is the identification of the common energy scale and the use of the nonlinear conductivity method.

The work will be of interest to researchers in the field of high-temperature superconductivity, as well as practitioners of the nonlinear conductivity technique.

- **Data & methodology:** Please comment on the validity of the approach, quality of the data and quality of presentation. Please note that we expect our reviewers to review all data, including any extended data and supplementary information. Is the reporting of data and methodology sufficiently detailed and transparent to enable reproducing the results?

The data appears to be of high quality and is presented clearly. I suggest a different colour or symbol for the LSCO $x=0.19$ data in fig 1b because it looks too similar to the grey squares in fig 1a. Some of the technical parameters of the model, such as the number of sites, would be useful to list to help readers reproduce the calculations.

We thank the reviewer for the thoughtful comments, which have motivated us to further strengthen our manuscript.

We have changed the colour of the LSCO $x = 0.19$ data. We note that the model is an effective medium calculation, and the schematic depiction (new Fig 2, old Fig. 3e) only serves as an illustration of the underlying physical picture. In order to provide a clearer description of the model in the main text, we have combined the prior Fig. 3e and Fig. S3 into a new Fig. 2. The details of the calculation are given in Methods.

- **Appropriate use of statistics and treatment of uncertainties:** All error bars should be defined in the corresponding figure legends; please comment if that's not the case. Please include in your report a specific comment on the appropriateness of any statistical tests, and the accuracy of the description of any error bars and probability values.

Uncertainties are reported and the sensitivity of the model calculations to different inputs is discussed.

- **Conclusions:** Do you find that the conclusions and data interpretation are robust, valid and reliable?

The data interpretation seems to be reliable. The disagreement with Ginzburg-Landau theory is evident.

- **Suggested improvements:** Please list additional experiments or data that could help strengthening the work in a revision.

The authors could comment on the role of critical fluctuations which are expected to be large in cuprates.

We have added a comment on critical fluctuations in the discussion on Page 9 and now also demonstrate that our model also captures the exponential tails observed in specific heat data (new Fig. 4e).

- **References:** Does this manuscript reference previous literature appropriately? If not, what references should be included or excluded?

Further evidence of pairing above T_c includes:

Photoemission work of Kondo et al., Nature Commun. 6, 7699 (2015).

Specific heat, Tallon et al., PRB 83, 092502 [2011].

Torque magnetometry, Kokanovic et al. PRB 88, 060505(R) [2013]

We thank the Reviewer for the suggested references, and have included them in the revised manuscript. As noted, we also have added a more extensive discussion of specific heat on Page 9 (including the reference suggested by the Reviewer, and two others), and mention the torque magnetization work on YBCO in the discussion as well.

- **Clarity and context:** Is the abstract clear, accessible? Are abstract, introduction and conclusions appropriate?

In the abstract I feel it is more appropriate to say that the emergence of superconductivity in the cuprates is "limited" by their intrinsic inhomogeneity, rather than "dominated" by it. Presumably the pairing itself still originates from interactions with some unidentified boson.

We agree that the pairing itself must originate from some boson exchange, but it seems to us that it is not clear what 'limited' would mean physically. We have thus changed this to 'strongly affected'.

Reviewer 2

In this work, Damjan Pele & co-workers reported experimental measurements of the nonlinear conductivity. The authors claimed the nonlinear conductivity vanishes exponentially above T_c , both with temperature and magnetic field, and exhibits temperature-scaling characterized by a nearly universal scale Ξ_0 , and that their findings are well captured by a simple percolation. The authors claimed that they have resolved a long-standing conundrum by showing that the emergence of superconductivity in the cuprates is dominated by their intrinsic inhomogeneity.

The phenomena associated with the pairing mechanism responsible for such unexpectedly high T_c in the HTSC have been subject to intense interest for many years, as they are important in terms of interesting, complex physics involved. The most important questions are: what causes superconductivity in the HTSC and crucially why are the transition temperatures so high? If these questions could be comprehensively answered then the pathway to develop new materials with even higher transition temperatures and other improved properties, such as higher critical current densities, would be clarified.

We agree, and emphasize three points: (i) Our manuscript introduces a novel experimental approach to the study of superconducting fluctuations, which we trust will be valuable in future studies of unconventional superconductors. We believe that this accomplishment alone renders our work highly significant and of broad interest; (ii) Irrespective of any interpretation, our finding of a *universal* pre-pairing regime in the cuprates is surely a significant advance in the field of unconventional superconductivity, and moreover a strong constraint on any theoretical understanding; (iii) The success of the percolation model strongly suggests that intrinsic inhomogeneity plays a vital role in cuprate physics, which again is a crucial insight toward understanding these complex materials and developing better high- T_c superconductors. These points are now further highlighted in the introductory part of the revised manuscript. We note that we now also demonstrate that our model can capture the exponential tails observed in specific heat measurements (see Fig. 4e and the corresponding text).

The main problem here is that all the demonstration rests on two unique samples for two representative cuprate families; a nearly optimally-doped sample of $\text{HgBa}_2\text{CuO}_{4+\delta}$ (Hg1201), an optimally-doped $\text{YBa}_2\text{Cu}_3\text{O}_{7-\delta}$ (YBCO) sample with 3% of Cu substituted by Zn (YBCO-Zn) and $\text{La}_{2-x}\text{Sr}_x\text{CuO}_4$ (LSCO) samples, (spanning a wide range of doping across the superconducting dome) are known as samples which exhibits considerable (La/Sr) point disorder.

YBCO has CuO chains. This makes YBCO different from other cuprates. The authors should consider more quantitatively the contributions of CuO chains on their experimental data. It is known that high quality YBCO samples (with sharp superconducting transition temperatures) have several different Cu-O chain superstructures: ortho II (O-II), O-V, O-VIII, O-III. It would be interesting to look the results for these samples.

First, we wish to emphasize that we have aimed to study as many diverse cuprate samples as possible, given the sample availability. Especially the considerable difference in (point) disorder between LSCO and Hg1201 is significant, and LSCO was used to study the phase diagram due to the relative ease of introducing hole doping in single crystals. Furthermore, we note that have gone to some length to compare our results to published measurements on other cuprates (old Fig. 3, new Fig. 4), including Eu-LSCO, underdoped Hg1201, and YBCO at several doping levels. Since the same generic behavior is observed, there is little room for doubt regarding the general validity of our conclusions. YBCO indeed has specific structural features, but due to sample unavailability, we are not in a position to do measurements on YBCO with chain superstructures; however, we had already included previous results

on YBCO, specifically paraconductivity measurements by Rullier-Albenque et al. in Fig. 4 (old Fig. 3). Moreover, in the revised manuscript, we further emphasize the specifics of YBCO and included a short discussion of torque magnetization results by the group of J. Cooper et al., ref. 15, where again the same universal exponential behavior is found. Finally, as noted, we now also demonstrate that our model can capture the exponential tails observed in specific heat measurements. This is specifically demonstrated for Ca-doped YBCO in the new Fig. 4e.

The open issues and novelty of the paper are not clearly presented. In the introduction, the authors mention several open questions and challenges concerning the subject of superconducting fluctuations in cuprates but the focus of the paper narrows down to the controversial issue of the validity of the Gaussian fluctuation picture and that their findings are well captured by a simple percolation.

In the revised manuscript, we have attempted to clarify that our work addresses the wider issue of large vs. small fluctuation regime (clearly showing that the temperature range is small), but very importantly also demonstrates non-Ginzburg-Landau behavior and remarkable universality. As noted, both the universal, unconventional temperature/field dependences and our successful modeling in terms of gap inhomogeneity have wide repercussions for cuprate physics.

The authors should more carefully select the references to previous studies. Their work follows several recent papers on the above controversy and many others that have not been cited. In fact, the controversy is old and experimental evidence of Gaussian fluctuations above T_c was reported by several groups [1-3].

[1] P.P. Freitas, C.C. Tsuei and T.S. Plaskett, Phys. Rev. B 36, 833 (1987).

[2] R. Hopfengartner, B. Hensel, and G. Saemann-Ischenko, Phys. Rev. B 44, 741 (1991).

[3] A. Gauzzi and D. Pavuna, Phys. Rev. B 51, 15420 (1995).

The Supplementary Information and Methods section includes a more detailed discussion of previous results, with focus on Ginzburg-Landau/Gaussian fluctuations. Ref. [2] actually already was cited in the Supplementary Information, and in the revised manuscript we now also include the other two references. Notably, references [1] and [3] do not find Gaussian behavior over an appreciable temperature range, especially further away from T_c (with Ref. [3] even claiming that the conventional power laws are not obeyed at any temperature).

The contribution of this paper for high- T_c superconductivity is not clear. From this result, what can the authors conclude? Is it the way to understand the origin of the pairing mechanism in the HTSC and to increase T_c ?

The experimental data are clear and the authors' analysis of their data may be appropriate. However, the results and discussion are highly specialized and are not suitable for general readers.

In conclusion, I don't think this work has enough novelty, contains new physics and is of significant importance to be published in Nature Communications.

The Reviewer seems to suggest that nothing less than a full solution of the cuprate high- T_c problem would be significant enough to warrant publication in Nature Communications, and it is certainly hard to

live up to such expectations. However, our finding of universal pre-pairing explained by intrinsic inhomogeneity in the cuprates constitutes a significant advance in the field of unconventional superconductivity, resolves long-standing questions, and demonstrates that our understanding of these materials must include inhomogeneity. While admittedly not providing a prescription for making a room-temperature superconductor, the present results do uncover this essential ingredient of cuprate physics, that might also play an important role in the superconducting mechanism itself. Importantly, as noted, the experimental technique that we introduce will be useful for studies of a wide range of unconventional superconductors. We believe that these are significant advances of broad interest that warrant publication in Nature Communications.

Reviewer 3

The present manuscript describes microwave resistance measurements on several hole-doped cuprates that claim to show a narrow region in temperature above the superconducting critical temperature in which superconducting fluctuations are present. The subsequent analysis shows that these fluctuations do not conform to the standard Ginzburg-Landau theory for critical fluctuations. On the basis of this analysis, the authors propose a percolation model for the development of superconductivity with decreasing temperature and introduce a new energy scale that is claimed to be universal and related to the distribution of (superconducting) gap magnitudes.

The debate regarding the nature of fluctuations in hole-doped cuprates has raged for a number of years, and in my opinion, this manuscript is a worthwhile contribution to this debate as it provides a different viewpoint that may help to guide future thinking. I am therefore minded to recommend its publication, provided that the Reviewers address the following points:

We thank the reviewer for the thoughtful comments, which have motivated us to further strengthen our manuscript.

(I) The data are not presented in a particularly transparent way. I could not deduce, for example, how the σ_3 data were extracted from the raw data. I recommend that the authors provide a couple of figures in the Supplementary Material explaining how the σ_3 plots shown in Figure 1 were obtained.

σ_3 was measured directly and independently from σ_1 , and the only step between raw data and Fig. 1 is a normalization to the value at the peak. Figure S1 shows the raw σ_3 data for representative samples, for direct comparison with Fig. 1.

(II) There is no plot of $P(T)$, i.e. the temperature dependence of the fraction of superconducting clusters. This is an important element in the overall story and its omission needs to be rectified.

We had stated that $P(T)$ is linear in the simplest possible scenario (Page 6) and discussed $P(T)$ in more detail in the Supplementary Information. A plot of the assumed $P(T)$ was provided in Fig. S3 (albeit with incorrect labels; this is now amended). In the revised manuscript, we have moved this figure to the main text and created a new Fig. 3 that also includes the schematic depiction of the percolating patches (previous Fig. 3e).

(III) Once the $P(T)$ plot is introduced, the authors should discuss how the evolution of $P(T)$ agrees (or not) with the typical behavior of the specific heat jump at T_c . Resistivity is of course a one-dimensional probe of superconductivity, while specific heat is a three-dimensional probe, and its temperature dependence through T_c should reflect to some degree the volume fraction of superconductor within the bulk of the samples.

This is a very good point; indeed, we find that the available specific heat measurements are (at least qualitatively) consistent with $P(T)$. In the revised manuscript, we include references to several specific heat investigations, and discuss them within the gap inhomogeneity scenario. We have also attempted to model the specific heat in the same simple manner as the tomographic density of states – by convoluting a mean-field specific heat jump with the gap distribution function. Tails in c_p above T_c are obtained, in

satisfactory agreement with experiment; this is now included in the new Fig. 4e, with details on the calculation given in the Methods section.

(IV) The discussion regarding the gap inhomogeneity deduced from STM is confined to Bi2212, which is not studied in the present manuscript. It is well known, e.g. from NMR linewidths, that this is the most inhomogeneous of the hole-doped cuprates. I would recommend the authors to add some qualitative discussion about the variation in inhomogeneity across the various families in the revised manuscript.

One of our main conclusions is that the inhomogeneity responsible for the observed effects is universal for different compounds, and thus at best weakly related to point disorder. We have expanded the discussion and now include a NMR reference for Bi2212 to better reflect this.

REVIEWERS' COMMENTS:

Reviewer #2 (Remarks to the Author):

The authors' have responded to most of my queries/comments in a satisfactory manner. I can recommend it for publication in Nature Communications.

Reviewer #3 (Remarks to the Author):

I am satisfied with the response of the authors to the previous round of refereeing and with the changes to the manuscript. I am therefore happy to recommend publication of the present manuscript in Nature Communications.